**Review**

# Radio-miRs: a comprehensive view of radioresistance-related microRNAs

Abraham Pedroza-Torres [ID],[1,2,*,†] Sandra L. Romero-Córdoba [ID],[3,4,*,†] Sarita Montaño [ID],[5] Oscar Peralta-Zaragoza,[6]
Dora Emma Vélez-Uriza,[7] Cristian Arriaga-Canon [ID],[8,9] Xiadani Guajardo-Barreto,[8] Diana Bautista-Sánchez,[10]
Rodrigo Sosa-León,[2] Olivia Hernández-González [ID],[11] José Díaz-Chávez,[8] Rosa María Alvarez-Gómez,[2] Luis A. Herrera[8,9]

[1]Programa Investigadoras e Investigadores por México, Consejo Nacional de Humanidades, Ciencias y Tecnologías, Mexico City C.P. 03940, Mexico
[2]Clínica de Cáncer Hereditario, Instituto Nacional de Cancerología, Mexico City C.P. 14080, Mexico
[3]Departamento de Medicina Genómica y Toxicología Ambiental, Instituto de Investigaciones Biomédicas, Universidad Nacional Autónoma de México, Mexico City C.P. 04510, Mexico
[4]Departamento de Bioquímica, Instituto Nacional de Ciencias Médicas y Nutrición "Salvador Zubirán", Mexico City C.P. 14080, Mexico
[5]Laboratorio de Bioinformática, Facultad de Ciencias Químico-Biológicas, Universidad Autónoma de Sinaloa (FCQB-UAS), Culiacán Rosales, Sinaloa C.P. 80030, Mexico
[6]Dirección de Infecciones Crónicas y Cáncer, Centro de Investigación Sobre Enfermedades Infecciosas, Instituto Nacional de Salud Pública, Cuernavaca, Morelos C.P. 62100, Mexico
[7]Laboratorio de Traducción y Cáncer, Instituto Nacional de Cancerología, Mexico City C.P. 14080, Mexico
[8]Unidad de Investigación Biomédica en Cáncer, Instituto Nacional de Cancerología–Instituto de Investigaciones Biomédicas–Universidad Nacional Autónoma de México (UNAM), Mexico City C.P. 14080, Mexico
[9]Tecnológico de Monterrey, Escuela de Medicina y Ciencias de la Salud, Monterrey, Nuevo León C.P. 64710, Mexico
[10]Department of Microbiology and Immunology, Life Sciences Institute, University of British Columbia, Vancouver, BC V6T 1Z3, Canada
[11]Laboratorio de Microscopia Electrónica, Instituto Nacional de Rehabilitación "Luis Guillermo Ibarraa Ibarra", Mexico City C.P. 14389, Mexico

*Corresponding author: Clínica de Cáncer Hereditario, Instituto Nacional de Cancerología, Av. San Fernando 22, Col. Belisario Domínguez Sección 16, Del. Tlalpan, Mexico City C.P. 14080, Mexico. Email: abraneet@gmail.com; *Corresponding author: Departamento de Medicina Genómica y Toxicología Ambiental, Instituto de Investigaciones Biomédicas, Universidad Nacional Autónoma de México, Circuito Mario de La Cueva s/n, C.U., Coyoacan, Mexico City C.P. 04510, Mexico. Email: sromero@iibiomedicas.unam.mx
[†]These authors contributed equally to this work.

Radiotherapy is a key treatment option for a wide variety of human tumors, employed either alone or alongside with other therapeutic interventions. Radiotherapy uses high-energy particles to destroy tumor cells, blocking their ability to divide and proliferate. The effectiveness of radiotherapy is due to genetic and epigenetic factors that determine how tumor cells respond to ionizing radiation. These factors contribute to the establishment of resistance to radiotherapy, which increases the risk of poor clinical prognosis of patients. Although the mechanisms by which tumor cells induce radioresistance are unclear, evidence points out several contributing factors including the overexpression of DNA repair systems, increased levels of reactive oxygen species, alterations in the tumor microenvironment, and enrichment of cancer stem cell populations. In this context, dysregulation of microRNAs or miRNAs, critical regulators of gene expression, may influence how tumors respond to radiation. There is increasing evidence that miRNAs may act as sensitizers or enhancers of radioresistance, regulating key processes such as the DNA damage response and the cell death signaling pathway. Furthermore, expression and activity of miRNAs have shown informative value in overcoming radiotherapy and long-term radiotoxicity, revealing their potential as biomarkers. In this review, we will discuss the molecular mechanisms associated with the response to radiotherapy and highlight the central role of miRNAs in regulating the molecular mechanisms responsible for cellular radioresistance. We will also review radio-miRs, radiotherapy-related miRNAs, either as sensitizers or enhancers of radioresistance that hold promise as biomarkers or pharmacological targets to sensitize radioresistant cells.

Keywords: microRNAs; radiotherapy; ionizing radiation; radioresistance; radio-miRs

## RT in oncology care

In cancer treatment, radiotherapy (RT) plays a key role in tumor cell control and is considered the most effective cytotoxic therapy for treating and shrinking solid tumors (Schaue and McBride 2015). RT uses ionizing radiation (IR) to inhibit and control malignant tumor cells' proliferation, growth, and metastasis (Wang *et al.* 2018). The success of RT is evidenced by its widespread use, as it is currently administered to approximately half of all cancer patients as neoadjuvant or adjuvant treatment, almost always with curative intent (Delaney *et al.* 2005).

RT can be applied as external beam RT, by implanting radioactive sources into cavities and tissues (brachytherapy) or by systemic delivery of radiopharmaceutical agents (Citrin 2017). Conventional external RT uses gamma ray photons from a radioactive decay source, or more commonly of X-rays produced by a linear accelerator, which generates megavoltage X-rays (Pereira *et al.* 2014). Several factors, including the type of radiation, dosing schedule, delivery technique, and the biological properties of the tumor and normal tissues, contribute to the clinical outcome after RT (Wang *et al.* 2018). The main challenge of radiotherapeutic

treatment is the emergence of radioresistance of tumor cells and associated toxicity. Many efforts have been made in this multidisciplinary field to improve the efficacy of RT to obtain better therapeutic results, minimizing the side effects associated with damage to adjacent normal tissues. Consequently, some patients may not respond favorably to this treatment and suffer recurrences that negatively affect their prognosis and quality of life. Therefore, it is necessary to identify resistance mechanisms and informative molecules to predict the efficacy of RT.

## Biological response to IR

RT uses IR to potently induce massive cell death by damaging the DNA of exposed cells, triggering the activation of death signaling pathways, and generating reactive oxygen species (ROS) and other reactive species associated with oxidative stress (Son *et al.* 2017; Wang *et al.* 2018; De Ruysscher *et al.* 2019). In addition, IR also produces DNA modifications in all cells exposed to the fractionated doses conventionally used (Srinivas *et al.* 2019).

There are multiple levels of multifaceted IR functions in various cellular contexts. At the cellular level, radiotherapeutic treatment causes morphological changes in the cell and multiple types of damages in subcellular structures and organelles, such as reorganization of the cytoskeleton, promoting changes in cell surface micromorphology (La Verde *et al.* 2021) and in mitochondria and endoplasmic reticulum (Kam and Banati 2013; Szumiel 2015; Mayer *et al.* 2016). At the intracellular level, IR modifies the amount and composition of negatively charged components within the plasma membrane (i.e. sialic acid, lectin, and calciumbinding sites), while cellular alterations include the formation of giant cells, deformations in nuclear morphology, and cells with multiple nuclei (Shadyro *et al.* 2002). Under these stressful cellular conditions established by RT treatment, cells activate compensatory survival pathways involving damage repair signaling, unfolded protein response, autophagy, and senescence (Nagelkerke *et al.* 2013; Biau *et al.* 2019; Huang and Zhou 2020).

Regarding the molecular level, IR mainly targets DNA molecules, causing double-stranded break or single-stranded break (SSB) damage, either directly or through the generation of ROS (Maynard *et al.* 2009; Giglia-Mari *et al.* 2011). Depending on the type of damage, the cell can induce diverse DNA damage response (DDR) mechanisms. Initially, IR activates several members of the phosphoinositide 3-related kinase (PI3K) family, such as ataxia telangiectasia mutant (ATM), Rad3-related protein (ATR), and DNA-dependent protein kinase. Although the functions of ATM and ATR are distinct, they coordinate many molecules that maintain the genomic integrity of the cell. SSBs and other DNA lesions, such as bulky DNA lesions and base mismatch errors, have been demonstrated to be efficiently repaired by several mechanisms, including base excision repair (BER), nucleotide excision repair (NER) (Hegde *et al.* 2008), and mismatch repair (MMR) (Hsieh and Yamane 2008).

However, when injury affects both DNA strands, the cell induces complex repair mechanisms involving homologous recombination (HR), nonhomologous end joining (NHEJ), and the recently described microhomology-mediated end joining (MMEJ) (Fig. 1). A detailed review of these mechanisms can be found in previous studies (Wang and Xu 2017; Huang and Zhou 2020).

Among the multiple molecules contributing to IR-related pathways, microRNAs (miRNAs) stand out for their pivotal role in IR-induced cell signaling events. Dysregulation of miRNAs following irradiation has been shown to be involved in the radiosensitivity of different cells and, consequently, in cell fate. miRNAs are small noncoding RNAs of approximately 19–22 nucleotide sequences that canonically exert their repressor effect on their target genes by binding to the 3′ untranslated region. However, they can apply their modulation through the 5′ end and the genebody sequence with different biological consequences (Cipolla 2014). As base pairing does not need to be perfectly complementary, a single miRNA can regulate hundreds of targets, while one target may be subject to regulation by many miRNAs (O'Brien *et al.* 2018). Therefore, these key posttranscriptional regulators influence the expression of relevant IR-related molecules necessary to establish radioresistance pathways (Galeaz *et al.* 2021). Dysregulation of miRNAs is a common event induced by DNA damage through the IR-activated ATM pathway that promotes KH-type Splicing Regulatory Protein phosphorylation. This protein stimulates the biogenesis of pri-miRNAs, leading to the synthesis of potential miRNA regulators of tumorigenesis (Zhang *et al.* 2011). Accumulating studies have demonstrated that miRNA upregulation can directly or indirectly influence DDR. For instance, PP1a gamma, an ATM agonist, is downregulated by miR-34a, thereby potentiating the DDR. In addition, negative feedback loops occur after DNA damage repair, such as miR-101, which is induced by ATM signaling (Rezaeian *et al.* 2020). Thus, the prior miRNA transcriptional landscape, as well as the subsequent miRNA profile established after irradiation, could influence the RT response.

## Molecular mechanisms associated with radioresistance and the role of miRNAs in this phenomenon

RT is one of the most relevant interventions for most human tumors, but neoplastic cells may not be controlled due to radioresistance, resulting in recurrence and metastasis. Some mechanisms related to tumor radioresistance are linked to DDR as this could impact several pathways governing basic functions within the cell, such as PI3K/AKT (also known as Protein kinase B, PKB)/mammalian target of rapamycin (mTOR) (linked to proliferation, cell survival, growth, and angiogenesis) (Xia and Xu 2015), ERK (linked to proliferation, differentiation, adhesion, migration, and survival) (Buscà *et al.* 2016), glycolysis (energy generation) (Bhatt *et al.* 2015), Vascular Enfothelial Growth Factor (angiogenesis) (Nieves *et al.* 2009), Notch (proliferation, differentiation, and apoptosis) (Lobry *et al.* 2011), and WNT/ß-catenin (proliferation, differentiation, migration, apoptosis, genetic stability, and stem cell renewal) (Pai *et al.* 2017).

Additionally, cancer cells that survive radiation exposure can develop a radioresistant phenotype by modulating various mechanisms and cell types, including autophagy, apoptosis, cell cycle control, ROS pathways, cancer stem cells (CSCs), and epithelial–mesenchymal transition (EMT), among others. It has also been found that miRNAs can act as a gatekeeper modulator in cellular signaling pathways that confer radioresistance. Based on the available preclinical and clinical data, we will discuss here the connection between miRNAs and the aforementioned modulatory mechanisms of radioresponse (Fig. 2; Supplementary Table 1).

### Autophagy and apoptosis

miRNAs have been associated with radioresistance through modulation of autophagy, although this phenomenon seems to be context- and tumor-dependent, as in some conditions, autophagy appears to confer resistance, while in others, it may enhance the response of cancer cells to RT (Mulcahy Levy and Thorburn 2020). For example, radioresistant pancreatic sublines derived from BxPC3 and PANC1 cells have lower levels of miR-23b and higher levels of autophagy (Wang *et al.* 2013). This is probably due to a decrease in ATG12, a known target of miR-23b, resulting

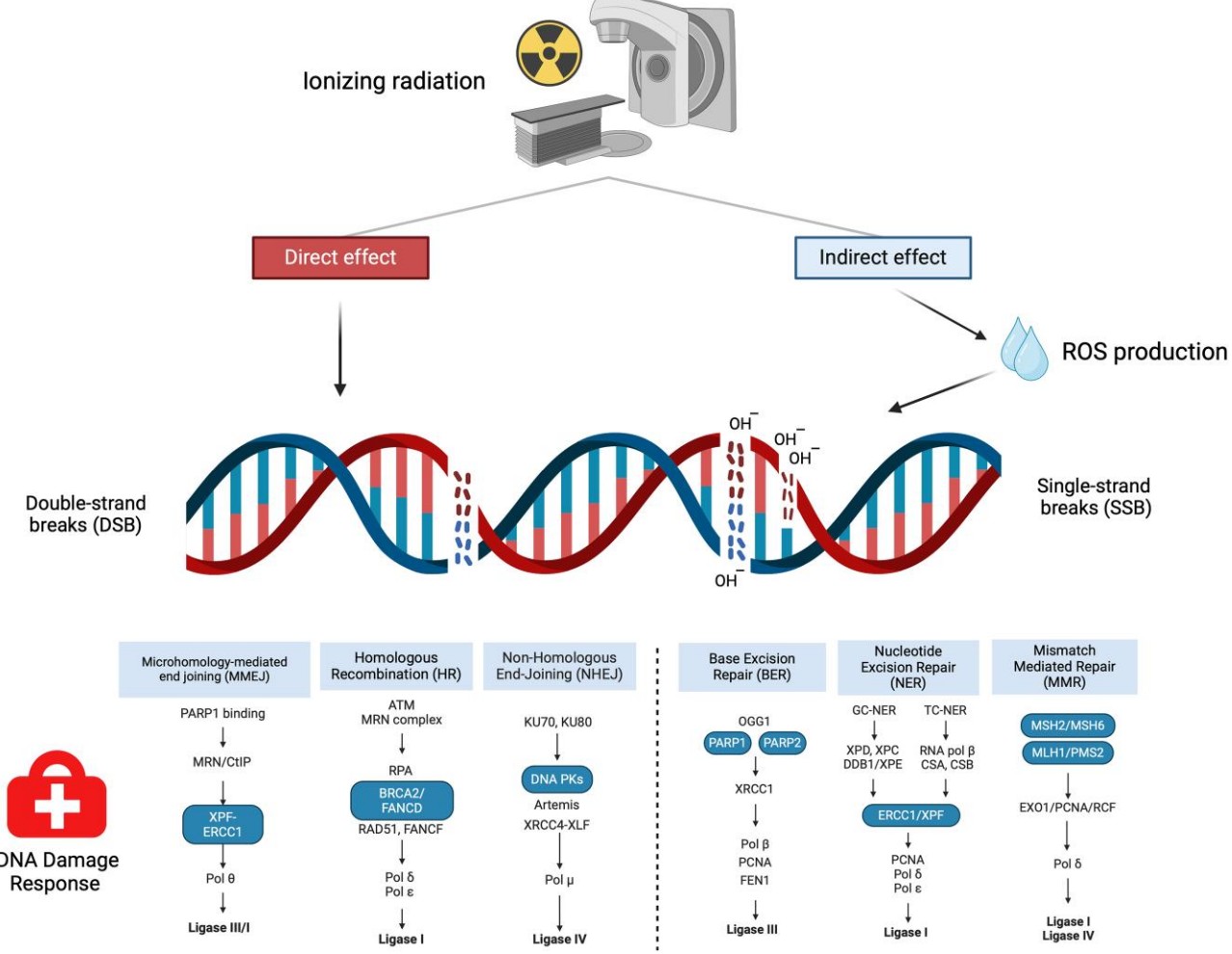

**Fig. 1.** DDR mechanisms. Cells respond to DNA damage caused by IR through various DDR mechanisms. Depending on the type of damage, cells can employ different repair pathways. For single-stranded damage, cells can utilize BER, NER, or MMR. When facing double-stranded DNA damage, they may opt for MMEJ, HR, or NHEJ mechanisms.

in radioresistance. Similarly, downmodulation of miR-216a, which suppresses BECN1 in a radioresistant pancreatic cancer cell line, enhances radiation-mediated autophagy and decreases radiation apoptosis (Zhang et al. 2015). In another study, miR-32 was shown to induce autophagy by targeting the autophagy inhibitor DAB2IP, improving the survival of prostate cancer cells after radiation treatment (Liao et al. 2015). Preclinical evaluation of 2 autophagy-inhibiting miRNAs has shown that overexpression of miR-26b, which regulates DRAM1 (Meng et al. 2018), and miR-200c, which downmodulates UBQLN1 (Sun et al. 2015), improves the sensitivity of cancer cells to RT. Meanwhile, the overexpression of miR-17 that negatively regulates ATG7, a key autophagy-promoting gene, reduces glioma cells sensitivity to radiation. miR-199a-5p plays a dual role as an autophagy modulator in breast cancer cell lines. In luminal MCF7 cells, miRNA suppresses IR-induced autophagy, while in basal MDA-MB231 cells, autophagy is induced, oppositely regulating DRAM1 and BECN1 genes in each scenario (Yi et al. 2013).

Apoptosis is a complex pathway influenced by miRNAs and regulates sensitivity to RT (Garcia-Barros et al. 2003). Some miRNAs have been shown to regulate the expression of antiapoptotic members of the B-cell lymphoma 2 family (Bcl-2). miR-193a-3p negatively regulates MCL1, and its exogenous expression in human cancer cell lines enhanced the IR-induced apoptotic response and cell growth inhibition (Kwon et al. 2013). Bcl-2 is a predicted target of miR-181a, and its ectopic expression in malignant glioma cells sensitized them to radiation treatment (Chen et al. 2010). miR-181a is also a target of PRKCD gene, a regulator of apoptosis, and its upmodulation inhibited radiation-induced apoptosis in cervical cancer cells, conferring radioresistance in vitro and in vivo (Ke et al. 2013). Expression of miR-24 inhibited cell growth in laryngeal squamous cell carcinoma models, enhancing apoptosis caused by radiation at different doses, probably by suppressing XIAP, an apoptotic suppressor gene (Xu et al. 2015).

## Cell cycle regulation

Different radiosensitivities have been observed in the cell cycle; late S-phase cells are the most radioresistant, while M-phase cells are the most radiosensitive. In fractionated RT, these cell cycle differences are considered, allowing tumor cells in a radioresistant phase to shift into a more radiosensitive phase (Syljuåsen 2019). Recent evidence has shown that miR-107 increases radiosensitivity in androgen-independent PC-3 prostate cancer cells through the upregulation of granulin, which, in turn, increases G1/S phase arrest and radiation-induced G2/M phase transition (Lo et al. 2020). Downregulation of miR-21 increases the in vitro radiosensitivity of colon, breast, and glioblastoma cancer cells by altering G2/M due to miRNA activity (Wang et al. 2009;

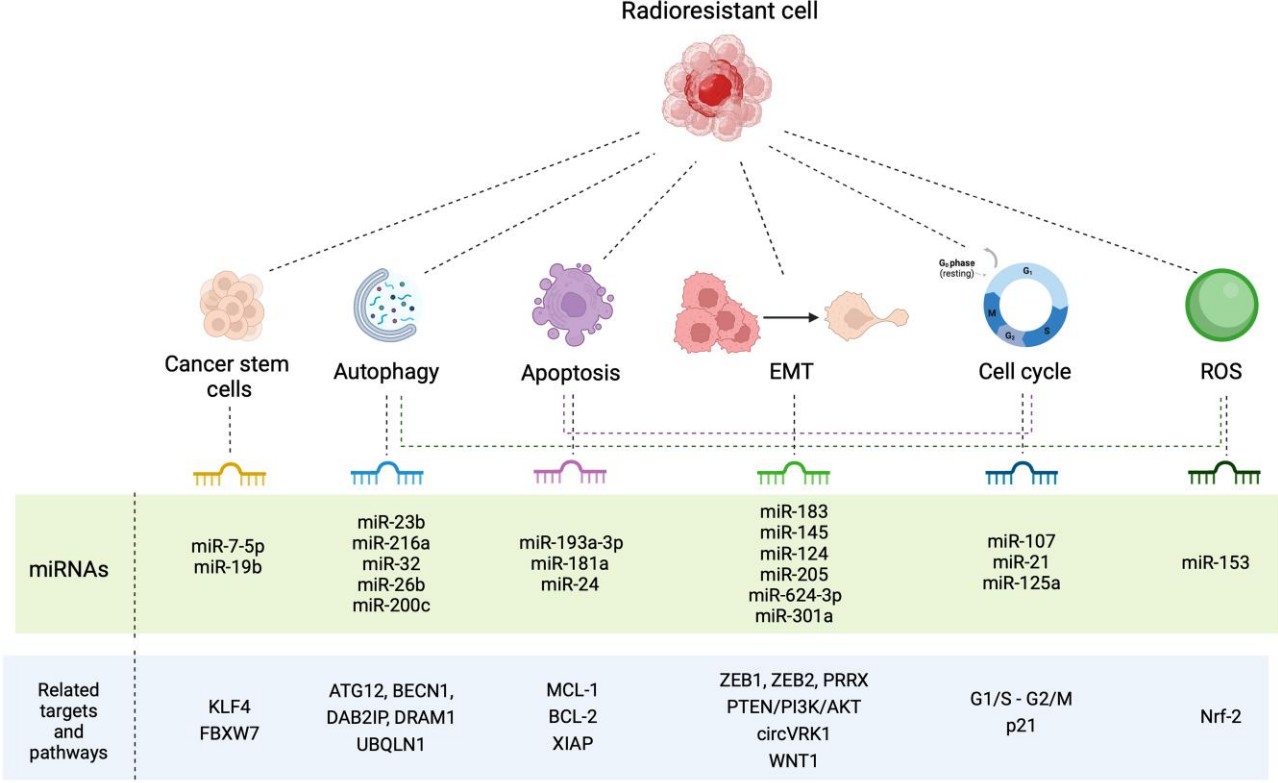

**Fig. 2.** Molecular mechanisms associated with radioresistance. Several molecular mechanisms and multiple molecular targets have been linked to the progress of radioresistance. These mechanisms include an elevated presence of CSC, increased autophagy, apoptosis dysregulation, EMT, cell cycle dysregulation, and an upsurge of ROS. In each of these instances, miRNAs capable of modulating the mechanisms responsible for the response to radiotherapy have been identified.

Anastasov *et al.* 2012). Data from our laboratory demonstrated that miR-125a modulates radioresistance through p21 in 3 radioresistant cervical cancer cell lines (SiHa, CaSki, and HeLa), and its exogenous expression in vitro sensitized the cell models to RT. miR-125a was also downregulated in radioresistant cervical cancer patients and radioresistant cell lines (Pedroza-Torres *et al.* 2018). There are tight regulatory and feedback mechanisms acting on DNA damage–related proteins that control the cell cycle, radiation sensitivity leading to cell death, and miRNAs.

## Formation of ROS

The local availability of molecular oxygen increases the effectiveness of RT, as ROS produced during water radiolysis can significantly damage DNA (Kawamura *et al.* 2018). Therefore, cancer cells with lower oxygen availability or higher consumption tend to be more radioresistant than those well-oxygenated cancer cells. For instance, the downmodulation of miR-153 in glioma CSC allows the upregulation of its putative target Nrf-2, which consequently leads to the accumulation of glutathione peroxidase 1. This antioxidant enzyme downregulates ROS levels, increasing radioresistance (Yang *et al.* 2015). Another example of ROS as regulators of important signaling pathways is the diverse transcriptional effect over stemness- and radioresistance-related miRNAs (miR-210, miR-10b, miR-182, miR-142, miR-221, miR-21, miR-93, and miR-15b) which varied depending on cell line subpopulation and clinicopathological features of Breast Cancer patients (Griñán-Lisón *et al.* 2020).

## Cancer stem cells

Apart from the intrinsic mechanisms of radioresistance, CSCs exhibit protective mechanisms that possibly lead to resistance to RT.

For example, lower levels of miR-7-5p, a stem cell–related miRNA, have been described in RT-resistant colorectal cancer cells. Exogenous expression of miR-7-5p in vitro and in patient-derived xenograft models led to the restoration of radiosensitivity by attenuating stem cell–like properties. In contrast, silencing of miR-7-5p in parental cells increased radiation resistance and enhanced CSC-like properties. In vitro experiments revealed the existence of an axis between miR-7-5p and KLF4, a stem cell–associated transcription factor that could contribute to regulating radiosensitivity in colorectal cancer cells (Shang *et al.* 2023). Further evidence also demonstrated that colorectal cancer–derived exosomes promote radioresistance and stem properties through the extracellular release of miR-19b, which is able to regulate FBXW7, a stem regulator (Sun *et al.* 2022).

## Epithelial–mesenchymal transition

It is a biological process that confers on the one side, invasiveness, and metastatic capacity to cancer cells and, on the other, radioresistance properties. Many mechanisms of radioresistance related to EMT have been described, and multiple miRNAs have been reported as modulators of EMT (Qiao *et al.* 2022). Among them, miR-183 was increased in radioresistant lung adenocarcinomas and negatively regulates the EMT promoter ZEB1 (Huang *et al.* 2021). Similarly, in human bladder cancer cells, miR-145 targets ZEB2, inhibiting EMT (Tan *et al.* 2015). Furthermore, miR-124 restoration could sensitize colorectal cancer cells to IR by directly targeting PRRX1, an EMT inducer (Zhang 2014). Given that PTEN/PI3K/AKT signaling also enhances radioresistance by promoting EMT, miR-205, a mechanistic regulator of this axis, could act as a brake on these phenotypes in esophageal squamous cell

carcinoma (Pan et al. 2017). Furthermore, miR-624-3p also modulates radioresistance by interacting with circVRK1, a circular RNA that mediates EMT (He et al. 2019). Another miRNA that increases radiosensitivity is miR-301a, which suppresses WNT1-dependent EMT (Su et al. 2019).

Altered miRNA transcriptional landscapes contribute to the development of radioresistance. However, they also arise as a consequence of radiation exposure in cancer cells, establishing intricate feedback loops between irradiated tumor cells and miRNA expression levels. This is because radioresistance occurs by intrinsic or acquired mechanisms. The former occurs when the cancer cell exhibits resistance signaling even before treatment begins and is somehow inherent to the biology of the tumor (West et al. 1998), while acquired radioresistance is the result of an adaptive process of cancer cells established by IR (Wu et al. 2023).

## TME and miRNAs as novel players in cancer radioresistance

For many years, most studies aimed at improving RT treatment outcomes have focused primarily on radiation-induced effects in the cancer cell compartment. However, as our understanding of tumors has evolved, tumor microenvironment (TME) components have begun to play a major role in delineating tumor resistance to RT. Many components of TME can render a tumor resistant to RT by intrinsic or de novo properties. Further, radiation therapy–mediated cell death can trigger its activity by enhancing the immune response against the tumor (Formenti and Demaria 2009), although under certain cellular contexts, antitumor immunity may be limited by the presence of radiation-resistant cells within the TME, such as myeloid and other suppressor effector cells, which consequently give rise to a radiation-resistant phenotype (Marigo et al. 2008; Deng et al. 2014; Fridman et al. 2017). Furthermore, despite the immunostimulatory effects of RT discussed above, it can promote an immunosuppressive TME in certain scenarios by recruiting myeloid-derived suppressor cell populations and repolarizing proinflammatory macrophages into anti-inflammatory ones (Vatner and Formenti 2015). On the basis of these points, it is rational to apply a combinatory treatment of RT (local effects) and immunotherapy (systemic effects), which in fact has been shown to outperform the therapeutic response (Theelen et al. 2021; Zhang et al. 2022).

miRNAs are master regulators of immune cell differentiation and function, as well as their interaction with other cell populations within the TME and cancer cells, and, when altered, promote immune evasion and the establishment of therapeutic resistance (Kousar et al. 2022). Increasing evidence points to miRNAs as key regulators in the immune response balance at tumor initiation and development, with potential clinical significance in immunotherapy. Several studies have pointed to various miRNAs as modulators of relevant features of cancer, such as inflammation, and as regulators of TME architecture (Contreras and Rao 2012; Marques-Rocha et al. 2015). Tumor cells and surrounding immune cells are also able to actively communicate through extracellular vesicles containing miRNAs, modifying immune and tumoral responses (Xing et al. 2021).

In the context of RT resistance, miR-21 increases PD-L1 expression, an immune checkpoint, by targeting PDCD4 and activating the PI3K/AKT pathway in breast cancer. Tumor models of miR-21 knock-in mice treated with anti-PD-L1 antibody and radiation resulted in increased T-cell, IFN-γ, IL-2, and reduced tumor size coupled to a decreased PD-L1 expression, thus improving immunotherapy via T-cell immune response activation (Guo et al.

2022). Further studies indicated that overexpression of miR-20b-5p in breast and lung cancer cells enhanced radiosensitization by repressing PD-L1/PD1. In vitro silencing of miR-20b-5p mimics the tumor-suppressing abilities of pembrolizumab (Jiang and Zou 2022). The overexpression of miR-378a-3p and its loading into exosomes were discovered by a further preclinical study focusing on exosomal miRNAs released from irradiated tumor cells and their in vitro characterization in cell line models and liquid blood biopsies of RT-treated mice xenograft model and patient samples. Granzyme B release by natural killer cells is decreased as a result of exosomal miR-378a-3p activity, contributing to tumoral immune escape (Briand et al. 2020).

As discussed above, miRNAs are important modulators of several resistance mechanisms, including autophagy. The role of miRNAs extends to complex regulatory networks of RT sensitivity, as predicted for autophagy-related miRNAs (miR-205-5p, miR-26a-1-3p, miR-6510-3p, miR-194-3p, miR-215-5p, miR-375-3p, miR-194-3p, miR-215-5p, and miR-375-3p) and miRNA–circular RNA (miR-194-3p/SESN3, miR-205-5p/ELAPOR1, and miR-26a-1-3p/SNCA) in patients with nonsmall cell lung cancer treated with RT. This axis has the potential to influence tumor immune status, as autophagy is associated with CD8 T-cell and gamma delta T-cell infiltration, while peroxisomal autophagy is associated with monocyte trafficking to tumors (Fan et al. 2021). EMT is also involved in intercellular communication between tumor cells and tumor-associated macrophages (TAMs) in colorectal cancer, promoting the antiproliferative polarization of TAMs through direct transfer of exosomes leading to a significant upregulation of miR-106b-5p. Mechanistically, miR-106b overexpression activates the PI3Kγ/AKT/mTOR cascade by directly suppressing PDCD4, enhancing EMT-mediated radioresistance, migration, invasion, and metastasis of tumor cells. In clinical samples, miR-106b was significantly increased and negatively correlated with PDCD4 expression levels, and exosomal miR-106b was enriched in plasma samples and significantly associated with poor progression (Zheng et al. 2015; Yang et al. 2021).

The cell communication and response to RT in glioblastoma are governed by TAM-derived extracellular vesicles that trigger the proneural–mesenchymal transition, a process responsible for progression due to increased resistance to RT. Small extracellular vesicles serve as a transfer mechanism for miR-27a-3p, miR-22-3p, and miR-221-3p from macrophages to tumor cells, promoting various mesenchymal phenotypes by targeting CHD7, which contributes to the maintenance of the proneural phenotype through the RelB/P50 and p-STAT3 pathways. Conditioning of a mouse xenograft model with small vesicles containing miR-27a-3p, miR-22-3p, and miR-221-3p enhanced radioresistance (Zhang, Xu, et al. 2020).

Despite major advances in our understanding of radioresistance and the miRNA-mediated radiation response, there is an urgent need to further explore the relationship between the response to RT, the tumor immune microenvironment, and regulatory molecules such as miRNAs. Overall, the above observations provide compelling evidence suggesting a functional interaction between miRNAs and immune signaling that influences radioresistance. This is important as combination therapies are being prioritized in the clinic.

## miRNAs as predictive biomarkers of tumor radioresponse

The important functional roles of these posttranscriptional regulatory networks have continually emerged, so we describe radio-miRs that have been described to act as radiosensitizers or enhancers of radioresistance and could be surrogate markers of local tumor

behavior reported in cohorts of patients with reproductive cancers (breast, cervical, and prostate neoplasms) and colorectal and lung neoplasms, all tumors treated with RT. Recent data highlight the potential of circulating miRNAs as new biomarkers to predict and monitor patient RT response. Circulating miRNAs are released by many cell types into various fluids, probably as endocrine and paracrine messengers that facilitate intercellular communication (Mi *et al.* 2013; Turchinovich *et al.* 2013), and are therefore becoming attractive noninvasive biomarkers that could be analyzed by liquid biopsy (Singh and Pollard 2017).

In developing countries, nearly 60% of prostate cancer patients undergo RT, indicated as first-line treatment for local and metastatic tumors as monotherapy or in combination with other interventions and administered as external beam radiation, brachytherapy, or radiopharmaceuticals including strontium-89, samarium-153, and radium-223 (Gay and Michalski 2018). Currently, RT for intermediate/high-risk localized tumors allows 5-year survival in 84% of patients, but resistance has still been observed (Widmark *et al.* 2019). For gynecological malignancies, including breast and cervical cancers, RT is a common treatment modality and may be used after breast surgery to reduce tumor recurrence or control metastases. In cervical cancer, radiation is widely used for locally advanced or recurrent tumors, improving local tumor control, disease-free time, and overall survival. Despite the successful use of RT in this type of cancer, some patients will still develop recurrences due to cell populations that exhibit intrinsic resistance or develop de novo resistance to RT, which increases cancer cell survival (de Souza Lawrence 2017). Thus, the identification of radiation response markers could be useful to monitor the evolution of RT and predict its therapeutic outcome (Table 1).

In patients with lung cancer, RT is an important treatment modality, either with curative or palliative intent. Unfortunately, some patients faced radioresistance that endangered their survival (Gomez-Casal *et al.* 2013; Willers *et al.* 2013). In addition, there is also substantial evidence of miRNAs related to radiosensitivity in lung cancer cells and their complex interaction with various biological processes and radiation-induced pathways that could help to gain insight into the sensitivity of tumors to irradiation (Table 2). Finally, the RT regimen, mainly in the preoperative setting, is one of the most important treatments for patients with colorectal cancer, but radioresistance is a crucial cause of poor therapeutic outcomes and one of the main problems in the management of this tumor (Duzova *et al.* 2021). The association of miRNA biomarkers with response to RT has been observed in some studies and could be used to determine radiosensitivity (Table 2).

Translating these radio-miR signatures into the clinic is an ongoing challenge, but importantly, the discussed data highlight the high informative value of radio-miRs, suggesting their potential use as molecular markers of response to RT. In addition to their prognostic value, radio-miRs could also provide information on radiosensitivity, leading to a new strategy to address treatment selection by predicting the efficacy of tumor cell response to IR. Further studies should be conducted to obtain robust and useful results in clinical trials and multicenter studies to demonstrate the association of miRNAs with radioresistance, which could impact clinical decisions in cancer patients, improving diagnostics and individualizing treatments. In conclusion, more information, basic research, and in vivo experiments are needed to identify circulating and local miRNA biomarkers that can be widely used as an approved RT response test.

## Understanding RT toxicity through radio-miRs

RT is a treatment that can affect not only tumor cells but also normal tissues exposed to irradiation, causing toxic effects. This can be associated with both short-term and long-term toxicity, encompassed as acute radiation syndrome. Short-term or acute toxicity refers to adverse effects that occur during therapy or within 3 months after RT; late effects appear thereafter and are usually considered irreversible and progressive over time. The adverse effects of radiation therapy toxicity are related to the target tissue and include cardiac toxicity, acute gastrointestinal damage, cognitive impairment, reproductive disorders, altered bone growth, hair loss, and secondary malignancy (Fig. 3) (De Ruysscher *et al.* 2019).

Finding molecular biomarkers to assess different levels of radiation toxicity is a crucial task in the study of radiation biomarkers. Early estimation of tissue damage due to radiation is fundamental to identify individuals who could potentially present toxicity and prevent this outcome in some way. In recent years and as already discussed, miRNAs have been widely reported as useful biomarkers in the field of IR, so new research lines are turning their attention to the potential of miRNAs as biomarkers in the prediction of radiation toxicity. Expression levels of local or circulating miRNAs can be used to predict radiation-induced damage, as these regulatory molecules are actively involved in the biological response to IR.

Consistent with this, miR-21, miR-146a, and miR-155 have been described as possible indicators of acute genitourinary radiotoxicity in prostate cancer patients measured in circulating peripheral blood mononuclear cells (Kopcalic *et al.* 2019). In a large cohort of prostate patients receiving intensity-modulated RT, miR-99a and miR-221 expressed in peripheral blood lymphocytes were found to be associated with genitourinary toxicity (Someya *et al.* 2018). Similarly, evaluation of miRNAs expressed in peripheral blood mononuclear cells from glioblastoma patients undergoing RT identifies miR-34a as a dosimeter of radiation exposure with variations in its expression over different time points. miR-10a is also associated with toxicity events and could predict the degree of toxicity, as well as miR-21 (Stepanović *et al.* 2022). Data from 63 patients with nonsmall cell lung cancer undergoing thoracic RT identified a serum miRNA signature (miR-100-5p, 106b-5p, 145-5p, 146a-5p, 192-5p, 195-5p, 223-3p, 25-3p, 34a-5p, 574-3p, 885-5p, Iet-7c, 200b-3p, and 134) capable of stratifying patients into high- and low-risk subgroups for developing cardiac toxicity due to RT (Hawkins *et al.* 2019). In this same type of cancer, miR-29a and miR-150 are related to the dose of radiation administered and may be useful in predicting the toxicity of thoracic RT (Dinh *et al.* 2016). In another study, the miR-6821 and miR-1290 expressions in the serum of patients with head and neck squamous cell carcinoma were associated with an increased risk of developing acute and late radiation toxicity (Higgins *et al.* 2017).

In addition to this evidence, the molecular basis for the association of miRNAs with radiotoxicity in breast cancer has also been described. For instance, RT increases the expression of miRNA-155, miRNA-221, miRNA-146, and miRNA-222 in patients who develop cardiovascular toxicity (Esplugas *et al.* 2019). In another study, miR-215 was also associated with cardiac toxicity due to RT. In a cohort of 86 patients, 24 circulating miRNAs, including miR-222, miR-328-3p, and miR-34a-5p, were identified differentially expressed between radiotoxicity groups and may serve as candidates for predicting radiotoxicity (Chalubinska-Fendler *et al.* 2017). Other studies have demonstrated the usefulness of urine in various urological cancers, such as prostate, bladder, and

**Table 1.** Reproductive cancers with highest incidence treated with radiotherapy.

| Cancer | Patients (total N) | Source | Radiotherapy | Associated miRNAs | Citation |
|---|---|---|---|---|---|
| | 149 | Plasma | Curative intent radiotherapy | miRNA-93 and miRNA-221 demonstrated a significant decrease in plasma levels after radiotherapy, suggesting a possible role of miRNA-93 and miRNA-221 as radiosensitizers in prostate cancer | Zedan et al. (2019) |
| | 30 | Tumor tissue | Neoadjuvant radiotherapy | miR-145 expression was significantly increased in patients demonstrating good response to neoadjuvant radiotherapy, while expression of the miR-145-regulated DNA repair genes was significantly decreased | Gong et al. (2015) |
| Prostate cancer | 33 | Normal vs tumor tissue (before and after RT) | Adjuvant radiotherapy | The expression of miR-541-3p was increased in tissues after radiotherapy, suggesting that its upmodulation after radiotherapy may affect radiosensitivity through regulation of the HSP27/β–catenin axis | He et al. (2021) |
| | 8 | Serum | Carbon ion radiotherapy | High expression of miR-493-5p, miR-323a-3p, miR-411-5p, miR-494-3p, miR-379-5p, miR-654-3p, miR-409-3p, miR-543, and miR-200c-3p before carbon ion radiotherapy predicted therapeutic benefit to RT. Similarly, post-RT expression of miR-654-3p and miR-379-5p is associated with radiotherapy efficacy | Yu et al. (2018) |
| | 25 | Serum | Curative radiotherapy | Exosomal expression of let-7a-5p and miR-21-5p was associated with radiation response | Malla et al. (2018) |
| | 16 | Blood leukocytes in 3 times: (1) prior RT; (2) dose reached of 2, 10, or 20 Gy; and (3) after therapy (46–50 Gy in total) | External beam radiotherapy | Overexposures to RT can affect normal tissues, and underexposures limit tumor control, so it would be useful to evaluate dosimetry methods in peripheral blood lymphocytes. miR-744-5p shows stable miRNA expression and, therefore, could serve as an informative miRNA to predict absorbed dose | Marczyk et al. (2021) |
| | 136 | Paraffin-embedded samples | Adjuvant radiotherapy | An inverse correlation between the expression of miR-200c–LINC02582 and CHK1 was observed, which might affect radiosensitivity | Wang et al. (2019) |
| Breast cancer | 20 | Formalin-fixed paraffin-embedded tumor | 45 Gy in 25 fractions plus a tumor bed boost of 16 Gy in 8 fractions | miR-139-5p is overexpressed in nonrelapsed patients possibly by miRNA regulation over multiple DNA repair and reactive oxygen species defense pathways | Pajic et al. (2018) |
| | 20 | Blood samples were collected from each patient at different times of the treatment | Hypofractionated RT (16 fractions, 2.65 Gy/ fraction) or conventional RT (25 fractions, 2 Gy/ fraction) | Identify 8 stemness- and radioresistance-related miRNAs (miR-210, miR-10b, miR-182, miR-142, miR-221, miR-21, miR-93, and miR-15b), which varied depending on clinicopathological features and across the pre-RT, during RT, and post-RT periods | Griñán-Lisón et al. (2020) |

(continued)

**Table 1.** (continued)

| Cancer | Patients (total N) | Source | Radiotherapy | Associated miRNAs | Citation |
|---|---|---|---|---|---|
| | 18 | Tumor tissue | Pelvic irradiation (45 Gy), parametrium boost of 10–14 Gy. Followed by intracavitary radiation therapy (20–25 Gy) | High levels of miR-181a related to RT resistance via silencing of PRKCD inhibiting irradiation-induced apoptosis | Ke *et al.* (2013) |
| | 30 | Tumor tissue | Conformal radiotherapy 2 Gy | miR-15a-3p is upregulated after radiotherapy, enhancing radiosensitivity by targeting tumor protein D52 | Wu *et al.* (2018b) |
| Cervical cancer | 41 | Tumor tissue | 55 Gy of radiotherapy and 30 Gy of internal brachytherapy | miR-31-3p, miR-3676, miR-125a-5p, miR-100-5p, miR-125b-5p, miR-200a-5p, and miR-342 were associated with clinical response and might inform radioresistance | |
| | 62 | Tumor tissue | 55 Gy of radiotherapy and 30 Gy of internal brachytherapy | miR-125a, which modulates CDKN1A, was downregulated in patients with cervical cancer who did not respond to standard treatment. | Pedroza-Torres *et al.* (2016, 2018) |
| | 53 | Tumor tissue | Preoperative radiotherapy | Low levels of miR-214-5p were detected in patients with poor radiotherapy response, which may be due to its regulation over ROCK1, which limits radiation sensitivity | Zhang *et al.* (2023) |

renal malignancies, as an auxiliary indicator to monitor the outcome of RT. Nephropathy is one of the common late effects in cancer survivors who received RT. miR-1224 and miR-21 were detected as late and acute toxicity markers, respectively, in urine samples from irradiated murine models and from human leukemia patients preconditioned with total body irradiation. In situ hybridization confirmed local expression of miR-1224 in mouse kidney tubes, and consistent with the mouse data, miR-1224-3p was overexpressed in human proximal renal tubular cells after irradiation (Bhayana *et al.* 2017).

Knowledge of the biological and molecular mechanisms of radiation toxicity offers new possibilities for preventing, mitigating, and treating the adverse effects of radiation therapy. A better understanding of the mechanisms of toxicity leads to a more rational approach to control it. Thus, in the next lines, we will review some of the advances reported on miRNAs' biological and mechanistic axes on radiotoxicity.

RT is an effective intervention in thoracic neoplasms but can cause lung injury and lead to respiratory failure. Mesenchymal stem cell–derived extracellular vesicles are now recognized as a therapeutic intervention for lung diseases as they have been shown to attenuate radiation-induced endothelial cell injury. One possible mechanism is by transferring miR-214-3p that downregulates ATM to minimize lung damage (Lei *et al.* 2021). miR-21 is another player observed in pulmonary fibrosis, as its expression contributes to collagen deposition through potentiation of TGF-β signaling (Kwon *et al.* 2016), whereas miR-29 represses type I collagen, reducing radiation-induced fibrosis through the negative modulation of the TGF-β/Smad signaling pathway (Yano *et al.* 2018).

Another side effect triggered by RT is vascular disease caused by chronic inflammatory activation that is partly orchestrated by irradiation-altered miRNAs. Low levels of miR-29b and overexpression of miR-146b have been observed in irradiated conduit arteries of patients undergoing microvascular reconstructions. Interestingly, exogenous in vitro and in vivo expression of miR-29b could dampen the vascular inflammatory response through inhibition of pentraxin-3 and dipeptidyl-peptidase-4 (Eken *et al.* 2019). The radiation-induced bystander effect is a toxic effect of RT that increases the risk of normal cellular damage after this treatment. Published evidence identified the role of miR-495 in alleviating this effect. Local irradiation of tumors overexpressing miR-495 resulted in fewer necrotic foci in nonirradiated liver tissue compared with controls, according to in vitro and in vivo tests (Fu *et al.* 2016).

Cranial RT to control central nervous system tumors could induce cognitive dysfunction. In this regard, human neural stem cell–derived exosomes prevent IR-induced cognitive dysfunction through a mechanism involving miR-124 transfer. Biological mechanistic evidence points to miRNA administration ameliorating cognitive dysfunction in cranially irradiated wild-type rodents (Leavitt *et al.* 2020). Other evidence points to miR-34a-5p as a promoter of cognitive deficits in total abdominal irradiated mouse models. miR-34a-5p targets Bdnf in the hippocampus and thus promotes cognitive dysfunction, which is alleviated after silencing the miRNAs (Cui *et al.* 2017).

It has also been reported that miRNA activity modulates the toxic effects of oral mucositis in the head and neck cancers. In this sense, diverse data reveal a novel mechanism involving miR-200c regulatory activity that suppresses the expression of proinflammatory cytokines (TGF-β, TNF-α, and IL-1α) by inhibiting NF-κB and Smad2 activation in normal human keratinocytes exposed to IR (Tao *et al.* 2019). Finally, miR-122-5p has been identified as exacerbating IR-induced rectal injury through its targeting of CCAR1. Notably, the in vivo knockdown of miR-122-5p following irradiation significantly alleviated IR-induced rectal injury in mice (Muhammad *et al.* 2021).

Although there are few studies on miRNA alteration during and after radiation therapy, mainly in human patients, future analyses will improve the understanding of how radiation toxicity occurs and how to detect it early using noninvasive methods to decrease its adverse effects. This is important, as the survival of cancer patients is increasing, and therefore, more patients are at risk of developing toxicity that can interfere with their survival and quality of life. miRNAs have been shown to have predictive potential for

**Table 2.** Colorectal and lung cancers commonly treated with radiotherapy.

| Cancer | Patients (total N) | Source | Radiotherapy | Associated miRNAs | Citation |
|---|---|---|---|---|---|
| | 19 | Fresh biopsies of rectal cancer tumors | Adjuvant pelvic locoregional radiotherapy (45–50.4 Gy) | Decreased miR-7-5p levels in cancer tissues of patients with rectal cancer resistant to radiotherapy, probably established by loss of miRNA modulation on the stem-associated transcription factor KLF4 | Shang et al. (2023) |
| Colorectal cancer | 106 | Plasma specimens from patients with advanced rectal cancer | Radiotherapy | 8-miRNA panel (miR-30e-5p, miR-33a-5p, miR-130a-5p, miR-210-3p, miR-214-3p, miR-320a, miR-338-3p, and miR-1260a) prioritized in pretreatment plasma specimens robustly predicted response to preoperative chemoradiotherapy | Wada et al. (2021) |
| | 60 | Fresh tumor tissue and blood samples | Radiotherapy | miR-25 is upregulated in both tumor and serum of radioresistant patients than in radiosensitive ones, whereas the putative target of the miRNA, BTG2, an antiproliferative gene, is downregulated, affecting sensitivity to radiotherapy | He et al. (2015) |
| | 30 | Tumor tissue | Postoperative radiotherapy (60 Gy) | In the radiotherapy-sensitive patients, 5 miRNAs (miRNA-126, let-7a, let-495, let-451, and let-128b) were significantly upregulated, and 7 (miRNA-130a, miRNA-106b, miRNA-19b, miRNA-22, miRNA-15b, miRNA-17-5p, and miRNA-21) were downregulated. miRNA-126 inhibited cell apoptosis through the PI3K-AKT pathway | Wang et al. (2011) |
| | 101 | Serum samples | Radiation (60–74 Gy) | High sera levels of miR-155 and miR-221 during the first 2 wk of chemoradiation therapy were associated with the development of severe radiation-induced esophageal toxicity | Xu et al. (2014) |
| | 5 | Platelet-depleted plasma over 5 dose points | Standard- or high-dose RT | The expression of miR-29a-3p and miR-150-5p in circulating exosomes decreased in response to radiation dose, which could help predict unexpected responses to radiotherapy | Dinh et al. (2016) |
| Lung cancer | 80 | Pretreatment serum sample | Standard- or high-dose RT | Dose–response score comprised of 11 miRNAs (miR-10b-5p, miR-125b-5p, miR-126-3p, miR-134, miR-155-5p, miR-200b-3p, miR-205-5p, miR-34a-5p, miR-92a-3p, miR-145-5p, and miR-22-3p) along with clinical factors to identify a subset of patients with locally advanced cancer who exhibit a clinical benefit, in terms of overall survival, from high-dose radiotherapy | Sun et al. (2018) |
| | 258 | Serum samples | Whole-brain RT (30 Gy) | Lower expression of miR-330 correlates with radiation sensitivity and poor prognosis in patients with brain metastasis from lung cancer, representing a novel predictor for radiation sensitivity and a novel therapeutic target for the prognosis | Jiang et al. (2017) |
| | 52 | Plasma samples | Radiotherapy | Plasma exosomal miR-96 is significantly overexpressed in the radioresistant group, and its expression has the potential to differentiate radioresistant from radiosensitive patients | Zheng et al. (2021) |

radiotoxicity in a liquid biopsy setting. Further studies are needed to identify miRNA biomarkers of radiation injury to minimize radiotoxicity. Thus, one of the ultimate goals of the researchers is to investigate further the mode of action of radioresponsive miRNAs, whose translation into biomarkers or intervention would be a novel strategy to maximize the safety of RT.

## Radio-miRs as cancer radiosensitizers: a novel translation to accelerate development of RT

Radiosensitizers are promising agents for enhancing RT tumor tissue injury by accelerating free radical production and DNA damage. In this section, we highlight some of the most promising macromolecules to be therapeutically administered as radiosensitizers. In the clinical setting, macromolecules such as miRNAs present relevant advantages in RT because they are found naturally in human cells and, therefore, have the cellular machinery for their processing and activity. Studies on miRNAs have elucidated several radiosensitization processes governed by miRNA regulation in multiple tumor types. Most of these miRNAs are downregulated in tumor tissues and radioresistant cells and are related to the regulation of DNA damage pathways. The use of RNA-based therapies, such as miRNA replacement therapy (miRNA mimics) or miRNA-encoding adeno-associated viruses,

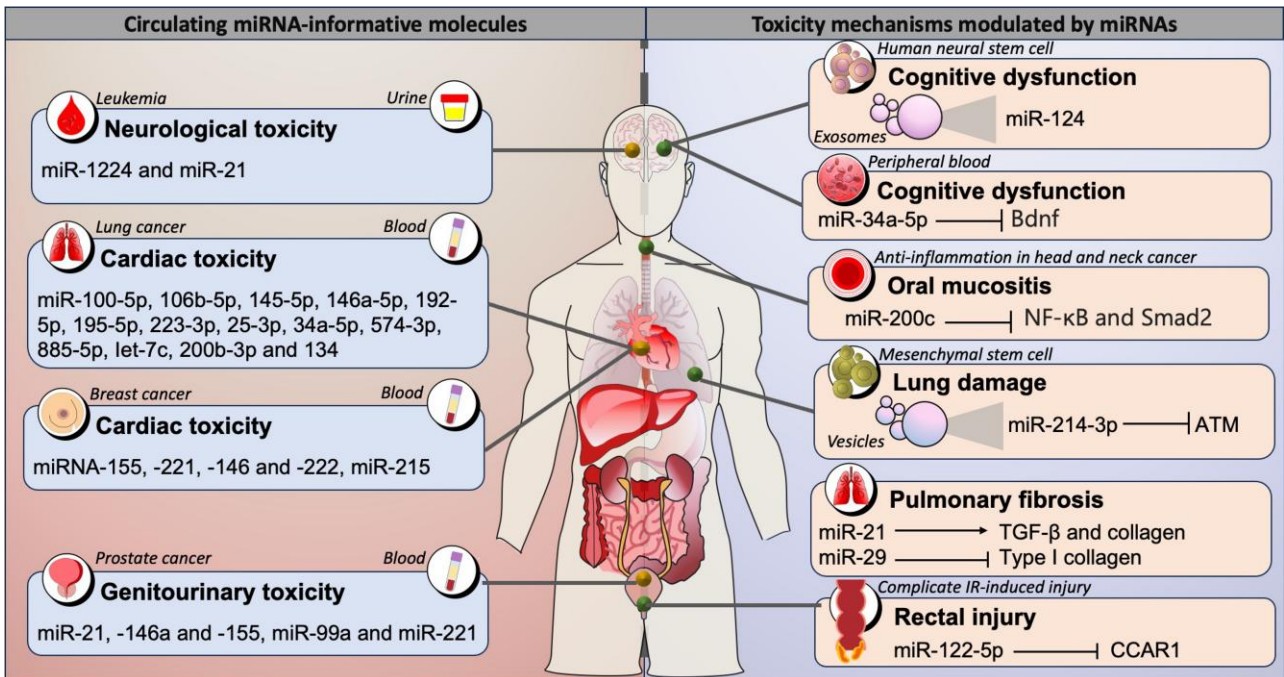

**Fig. 3.** Circulating miRNA-informative molecules and toxicity mechanisms modulated by miRNAs. Radiotherapy can cause both short- and long-term toxicity, causing various adverse effects such as cardiac toxicity, acute gastrointestinal damage, cognitive impairment, reproductive disorders, bone growth disturbances, hair loss, and secondary malignancies. In these processes, various miRNAs have been identified as molecular information on IR toxicity, such as miR-124 and miR-34a-5p in cognitive dysfunction, miR-214-3p in lung damage, and miR-21 and miR-29 associated with pulmonary fibrosis.

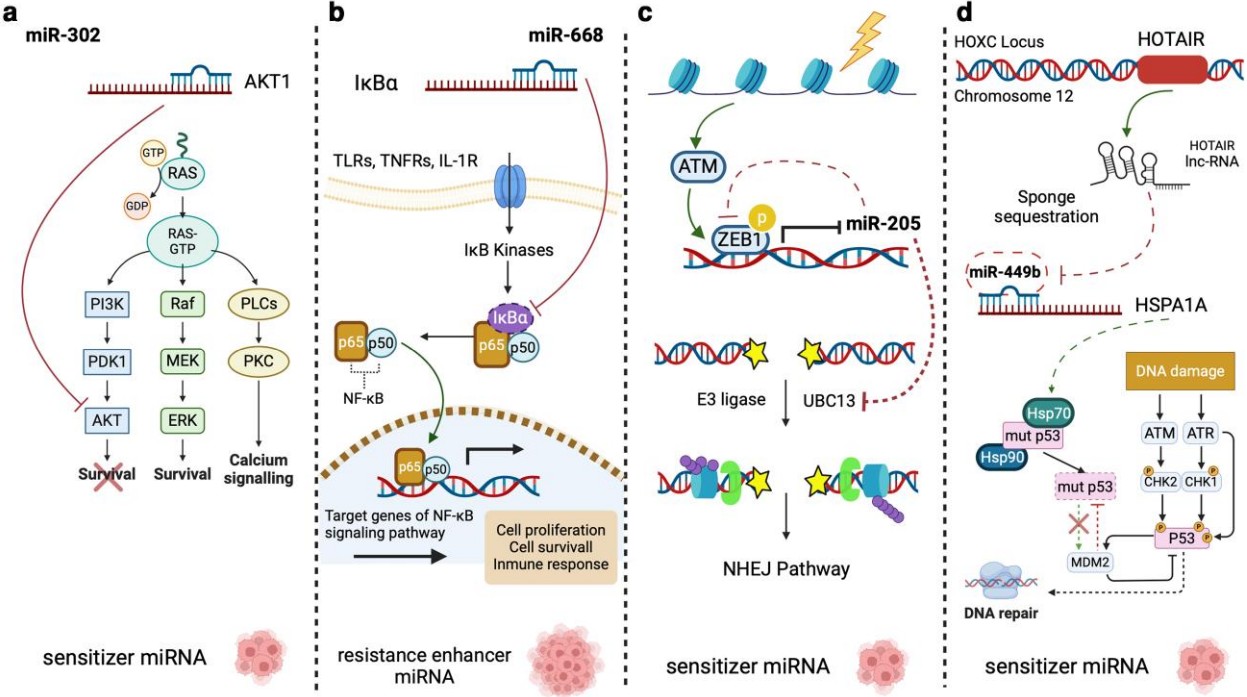

**Fig. 4.** Different mechanisms of radio-miRs for modulating the radioresistant phenotype. Radio-miRNAs employ various mechanisms to regulate critical cellular targets involved in the modulation of radioresistance. Some of these mechanisms involve direct regulation of targets associated with radioresistance, such as miR-302 and AKT/RAD52 a). Another common approach is indirect regulation through intermediaries within the proteins associated with radioresistance processes, like the miR-669–NF-κB signaling pathway b). Furthermore, radio-miRNAs participate in complex cellular regulatory networks that allow them to modify the radioresistant phenotype, as seen with miR-205 interaction with ATM c) and the lncRNA HOTAIR's role in miR-449 regulation d).

should be a therapeutic approach to restore miRNA expression with the goal of improving RT outcomes.

Radio-miRs exhibit different mechanisms for modulating the radioresistant phenotype of cells (Fig. 4). For instance, mimicking diverse miRNA expression results in strong enhancement of RT due to significantly increased of oxidative stress and accumulation of unrepaired DNA damage, preventing DNA stress tolerance (Kofman *et al.* 2013; Zhang, Wang, *et al.* 2014; Pajic *et al.* 2018; Song *et al.* 2018). A gatekeeper gene of the DNA damage repair process is p53-regulating signaling pathways such as AKT/mTOR axis (Qu *et al.* 2015; Chang *et al.* 2016; Wu *et al.* 2018a; Shao *et al.* 2019). Another relevant mechanism involves the direct regulation of cellular targets crucial in radioresistance modulation (Liang *et al.* 2013). Other recurrent mode of action involves the indirect regulation through intermediates of proteins involved in radioresistance processes (Luo *et al.* 2017).

In addition, radio-miRs participate in intricate cellular regulatory networks that allow them to modify the radioresistant phenotype by intervening in cross-regulatory pathways such as ATM/ZEB1/miR-205 (Zhang, Wang, *et al.* 2014; Rezaeian *et al.* 2020). These interactions could also occur with other noncoding regulatory RNAs such as long noncoding RNAs (lncRNAs), which can act as a miRNA sponge, effectively sequestering miRNAs and thereby facilitating the expression of their target genes (Zhang, Wang, *et al.* 2020).

There are multiple pieces of evidence for miRNA activity in the regulation of RT in oncologic diseases. To provide an in-depth characterization of these data, we present a digital pivot table (Supplementary Table 1) that provides summary information on the reported mechanism or informative value of 157 miRNAs in 38 biological scenarios, associated with 22 RT responses discussed in 165 articles.

Exogenous therapeutic expression of miRNAs remains a challenge, but in recent years, substantial efforts have been made toward this clinical application. The preclinical evidence presented here and growing knowledge of miRNA-DDR pathways demonstrate the potential for therapeutic modulation in oncology. Exploring miRNAs related to tumor radiosensitivity is emerging as an effective strategy for the development of radiosensitizers, reinforcing their putative translational impact as a modulator of genomic instability.

## Conclusion and future perspectives

Today, clinical cancer care includes multidisciplinary approaches to ensure the maximum benefit of each patient to treatment. RT is an effective cancer intervention, and currently, almost 50% of all cancer patients receive it during the course of their disease, alone or in combination with other therapeutic options. Unfortunately, although radiation treatment is effective for a huge number of patients, a number of them do not have a clear clinical benefit.

In a biology-driven era, RT has benefited from new avenues focused on the study of the biological activity of relevant molecules in RT toward a precision medicine strategy. The molecular landscape of certain cell populations could provide advantages in adapting to RT-induced damage and developing resistance mechanisms that represent a latent and relevant obstacle in cancer treatment. Evidence has shown that miRNAs play a crucial role in regulating the response to IR across various tumor cells, by modulating the expression of key genes in cellular processes associated with the development of radioresistance. These miRNAs, here termed radio-miRs, can act as radioresistance or radiosensitizing miRNAs. It is also well known that malignant lesions are formed not only by cancer cells but also by a wide variety of other cells responsible for the formation of the TME. Recently, it has become clear that environmental factors

interact with each other to create a complex ecosystem that either enhances or diminishes the effects of radiation. In this context, miRNAs also play a key role as modulators and communicators. Although, to date, most studies on the biology of miRNA regulation remain correlative, we believe that the availability of new technologies and molecular characterization of human samples will provide the opportunity to delve deeper into the functional mechanics of miRNA phenotypes.

Despite significant advances in understanding the mechanisms of radioresistance, informative molecules to predict clinical and biological outcome remain limited. Relatively recent studies have found that radio-miRs may have value as local or circulating biomarkers, opening the door to their future translation to the clinic as feasible biomarkers to predict and monitor clinical response to RT, improving the ability to diagnose and individualize treatments in cancer patients. Although miRNAs have not yet been confirmed and licensed for cancer treatment, their clinical applicability can also be explored by targeting actionable oncology molecules and pathways to improve response to RT. In addition, careful consideration must be given to the methods for selectively delivering therapeutic miRNAs to specific cells. This aspect poses a significant challenge, marking a crucial step in the advancement of miRNA-based therapeutic molecules. One potential solution to this challenge involves the development of aptamer–miRNA conjugates. These conjugates possess distinctive characteristics that facilitate the targeted delivery of therapeutic miRNAs to specific cells, achieved through the recognition of cellular receptors, particularly in tumor cells. Nevertheless, there is a need for further optimization of this technique, including determining the appropriate tumor cell receptor and optimization of concentrations for the effective delivery of therapeutic miRNAs.

Finally, advances in data science will allow the application of artificial intelligence models to miRNA omic data characterized in the field of RT, which will allow the improvement of RT response and the development of personalized treatments for patients.

## Data availability

An in-depth characterization of the radio-miRs involved in modulating the cellular response to radiation is presented in the form of a pivot table available digitally in the supplementary material.

Supplemental material available at GENETICS online.

## Conflicts of interest

The authors declare that the research was conducted in the absence of any commercial or financial relationships that could be construed as a potential conflict of interest.

## Author contributions

APT and SLRC conceived and designed the study. SM, OPZ, DEVU, CAC, XGB, DBS, OHG, and RSL did the literature research and data acquisition. APT and SLRC design and draw the figures. JDC, RMAG, and LAH provided insightful thoughts to study concepts and discuss the manuscript. All authors wrote, read, and approved the manuscript.

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
