## [Peer Review File · Genetics]

Radio-miRs: a comprehensive view of radioresistance-related microRNAs

Abraham Pedroza-Torres, Sandra L. Romero-Córdoba, Sarita Montaña, Oscar Peralta-Zaragoza, Dora Vélez-Uriza, Cristian Arriaga-Canon, Xiadani Guajardo-Barreto, Diana Bautista-Sánchez, Rodrigo Sosa-León, Olivia Hernández-González, José Díaz-Chávez, Rosa María Álvarez-Gómez, and Luis A. Herrera

NOTE: The reviews and decision letters are unedited and appear as submitted by the reviewers.

In extremely rare instances and as determined by a Senior Editor or the EIC, portions of a review may be redacted. If a review is signed, the reviewer has agreed to no longer remain anonymous.

The review history appears in chronological order.

Review Timeline:

Submission Date:	2023-11-16
Editorial Decision:	2024-01-29
Resubmission Received:	2024-04-02
Editorial Decision:	2024-04-24
Revision Received:	2024-05-21
Accepted:	2024-05-29

January 29, 2024

GENETICS-2023-306614

Radio-miRs: a comprehensive view of radioresistance-related microRNAs

Dear Dr. Pedroza-Torres:

Two experts in the field have reviewed your manuscript, and I have read it as well. I apologize for the length of time that has passed since submission. We had a difficult time finding reviewers, particularly at this time of year. However, I am pleased to inform you that, with minor revisions, it is potentially suitable for publication in GENETICS. The reviewers have comments and concerns that need to be addressed in a revised manuscript. You can read their reviews at the end of this email.

I think you should be able to address all the comments of both reviewers. They each have a single request for more information that I think you should be able to provide, and there are also some suggestions from Reviewer #2 for relatively simple condensing and reorganization. Both reviewers also list typographical errors that should be corrected.

We look forward to receiving your revised manuscript. Please let the editorial office know approximately how long you expect to need for revisions.

Upon resubmission, please include:

1. A clean version of your manuscript;
2. A marked version of your manuscript in which you highlight significant revisions carried out in response to the major points raised by the editor/reviewers (track changes is acceptable if preferred);
3. A detailed response to the editor's/reviewers' comments and to the concerns listed above. Please reference line numbers in this response to aid the editors.

Additionally, please ensure that your resubmission is formatted for GENETICS.

<https://academic.oup.com/genetics/pages/general-instructions>

Follow this link to submit the revised manuscript: Link Not Available

Sincerely,

Jeff Sekelsky
Senior Editor
GENETICS

Approved by:
Howard Lipshitz
Editor in Chief
GENETICS

Reviewer #1 (Comments for the Authors (Required)):

GENETICS-2023-306614

Radio-miRs: a comprehensive view of radioresistance-related 1 microRNAs

Pedroza-Torres et al.

This report provides a comprehensive overview of radio-miRs and their roles in radiotherapy response, as prognostic biomarkers, and as potential interventional tools. There are a number of minor issues that need to be addressed, listed below. In terms of intervention, one significant gap in the presentation relates to how the authors envision differential miR effects in tumor/TME vs normal tissue: how would miR treatment be limited to just tumor tissue (if radiosensitizing) or to just normal tissue (if radio-protecting)? Some insight into this question (possible selective interventions and associated challenges) can be added to the final perspectives section.

Minor issues and typos

The cited references need to be formatted fully in English.

In a few places the authors use the terms radioresistance and radio-potiation, apparently as contrasting concepts. Do the authors mean to contrast radioresistance with radio-protection? The term potentiation in these contexts is unclear - if this is the intended term, it should be clearly defined.

Headings in Tables 1 and 2: "Associated miRNAs" misspelled.

Line 70: ...evidence suggests that overexpression of DNA repair systems, increased reactive oxygen species, tumor microenvironment, and enrichment of cancer stem cell populations can be considered the main events of radioresistance.

This should be rephrased. I understand what the authors intent is here, but this list doesn't describe 'events' so much as 'factors'. For example, TME is not an event.

Line 73: ...(miRNAs), master regulators of gene expression.... I agree that miRs are important regulators of expression, but 'master' is an overstatement, as promoters and transcription factors are at least as important as miRs. Better to state that miRs are key or important regulators.

Line 98: ...all with curative intent. This should be 'most' or 'almost always' with curative intent as RT is used in some cases for palliative treatment, i.e., whole brain irradiation to mitigate pain.

Lines 135-141 and Fig. 1: IR induces both DSBs and SSBs. Also ROS generated by IR also induces both SSBs and DSBs. Fig 1 implies that IR induces DSBs directly while indirect effects (through ROS) induce single-strand damage - this is incorrect: both direct energy absorption and indirect effects of ROS can yield both double- and single-strand damage, depending on the spatial distribution of the energy deposition/ROS actions This paragraph and figure need to present these facts more accurately.

Line 147: SSBs can be effectively repaired by base excision repair (BER), nucleotide 147 excision repair (NER) and DNA mismatch repair (MMR) mechanisms.

This is technically incorrect - SSBs are indeed created by these repair systems, so in the end SSBs are managed by these repair systems. However, SSBs themselves can be simply repaired by DNA ligation.

Line 157: IR doesn't cause premature chromosome condensation - PCC is instead a technique used to identify chromosomal aberrations after IR.

Line 201: unclear as written. Rephrase as: ...through modulation of mechanisms such as autophagy...

Line 229: IR-induced (?)

Line 288: ...CircVRK1, an EMT-mediated signaling molecule.

Line 301: is there a more up-to-date reference for acquired resistance than ref 68 (from 1967)?

Lines 345-350: this complex sentence seems to be missing a phrase - note the unpaired parenthesis in the middle.

Line 353: ...resistance mechanism...

Line 385-387: a reference supporting this statement should be added.

Line 426: do you mean ionizing radiation instead of infrared?

Line 434: RT may have systemic effects, but it is highly targeted so it is odd to describe as a systemic treatment, like chemotherapy.

Line 524: Rephrase: In vivo knock-down of CCAR1 after irradiation....

Line 534: radio-responsive miRNAs

Line 569-570: should be written as ATK and RAD52 instead of AKT/RAD52 as the slash designation usually indicates two names for the same molecule or gene.

Reviewer #2 (Comments for the Authors (Required)):

In the review entitled "Radio-miRs: a comprehensive view of radioresistance-related microRNAs," Pedroza-Torres, Romero-Cordoba, and colleagues synthesize an abundance of information relating to the expression and involvement of miRNAs in cancerous tissues and cells that are undergoing or have previously experienced radiation therapy. They discuss the potential for miRNAs to influence resistance or sensitivity to radiotherapy, and speculate about which miRNAs could be targets for therapeutic approaches and/or used as biomarkers for radiotherapy effectiveness. Overall, this review entails a large amount of work. The figures are helpful, detailed, and will be of benefit to the field. They serve as a nice summary of the information provided in the review. However, there are some improvements that could be made to the review to clarify several points and improve its utility across fields.

First, I found that, as written, the linkage and rationale between radiotherapy and miRNAs was somewhat brief and not necessarily well-connected. I understand the interest of the authors in this topic, but think that it would be useful to lay out the rationale more explicitly and clearly. This could be done by reframing the abstract and introduction, pointing out the "big picture" means by which miRNAs could modulate responses to radiotherapy, and building on these major themes throughout the rest of the review. This is done to some extent, but it needs to be more systematic. For example, lines 116-121 and 163-166 are the rationale/linkage between radiotherapy and miRNAs, but they are somewhat weak. For instance, could we not also look at transcription factors or some other class of gene expression regulators-I wanted to know why miRNAs, specifically, in radiotherapy?

Second, the abstract and review are wordy and could be streamlined. One way to do this would be to break it up into multiple reviews (miRNAs as markers is somewhat a separate topic). Another way is to simply streamline the writing style to make it more succinct-a lot of the early part of the review is dedicated to radiotherapy cellular responses, which is necessary, but could be shortened to get into the discussion of miRNAs (consistent with the title of the review) at an earlier point in the review.

Third, the figures very nicely summarize much of the writing in a more intuitive way. In contrast, in many sections the text read like a laundry list of miRNAs, with a one-sentence summary of what a miR does in cancer tissue/type X/Y/Z. I found the format of statements like "miR A does X in tissue Y" difficult to keep in focus and really absorb as I read, and would have benefitted from having a more unified discussion that placed the biological role at the forefront and then detailed related miRNAs or related cancers or used some other common biology thread to link the discussion. This was done to some extent in the molecular mechanisms section and subsections, and I would have benefitted from more of this type of organization throughout the review.

Line 115: delete "that provide helpful information"

Line 154 "multiple damages in" should be "multiple types of damage"

Line 161: clarify the difference between multiple cell nuclei and cells with numerous micronuclei

Line 210: "sublines" should be "sub-lines"

Line 249: "down-modulated" should be "down-regulated"

Line 265: "Cancer stem cells:" should be bold

Line 404 "disease-free" should be "disease-free time"

Author's response

Dear Reviewers,

We sincerely appreciate the time and effort you dedicated to reviewing our work and your valuable and constructive comments and suggestions. We have addressed all of your comments and recommendations.

Point-by-point response to Reviewer 1's comments.

1.- In terms of intervention, one significant gap in the presentation relates to how the authors envision differential miR effects in tumor/TME vs normal tissue: how would miR treatment be limited to just tumor tissue (if radiosensitizing) or to just normal tissue (if radio-protecting)? Some insight into this question (possible selective interventions and associated challenges) can be added to the final perspectives section.

Response:

This point represents one of the main challenges in utilizing miRNAs as therapeutic molecules. Nevertheless, various approaches have been proposed as potential solutions to this issue (PMID: 24881764, PMID: 31855835, PMID: 33294587, PMID: 34236760). This information has been incorporated into the Conclusion and Future Perspectives section.

2.- The cited references need to be formatted fully in English.

Response:

We have carefully reviewed all references and have modified those that were not entirely in English.

3.- In a few places, the authors use the terms radioresistance and radio-potential, apparently as contrasting concepts. Do the authors mean to contrast radioresistance with radio-protection? The term potential in these contexts is unclear - if this is the intended term, it should be clearly defined.

Response:

We apologize for any confusion. To enhance clarity within the article, we have standardized the terminology and replaced the aforementioned phrase with 'radiosensitizers' and 'enhancers of radioresistance.

4.- Headings in Tables 1 and 2: "Associated miRNAs" misspelled.

Response:

Both headings have been reviewed and corrected.

5.- Line 70: ...evidence suggests that overexpression of DNA repair systems, increased reactive oxygen species, tumor microenvironment, and enrichment of cancer stem cell populations can be considered the main events of radioresistance.

This should be rephrased. I understand what the author's intent is here, but this list doesn't describe 'events' so much as 'factors'. For example, TME is not an event.

Response:

Thank you for your suggestion. In the revised version of the manuscript (lines 61-64), we acknowledge that the original list does not accurately describe events, but rather factors contributing to radioresistance. We have revised the wording to reflect this more accurately. The sentence now indicates that overexpression of DNA repair systems, increased reactive oxygen species, modulation of the tumor microenvironment, and enrichment of cancer stem cell populations are key factors contributing to radioresistance. We believe this revision enhances the clarity and accuracy of the text.

6.- Line 73: ...(miRNAs), master regulators of gene expression.... I agree that miRs are important regulators of expression, but 'master' is an overstatement, as promoters and transcription factors are at least as important as miRs. Better to state that miRs are key or important regulators.

Response:

We have revised the mentioned line of text. The updated version now utilizes 'Key Regulators' (lines 63 -64) in the new version of the manuscript. This version maintains clarity and conciseness while addressing the adjustment suggested.

7.- Line 98: ...all with curative intent. This should be 'most' or 'almost always' with curative intent as RT is used in some cases for palliative treatment, i.e., whole brain irradiation to mitigate pain.

Response:

Thank you for your comment. We have considered the aforementioned therapeutic options of radiotherapy and have corrected the text by incorporating the phrase 'almost always' with curative intent (line 90).

8.- Lines 135-141 and Fig. 1: IR induces both DSBs and SSBs. Also, ROS generated by IR also induces both SSBs and DSBs. Fig 1 implies that IR induces DSBs directly while indirect effects (through ROS) induce single-strand damage - this is incorrect: both direct energy absorption and indirect effects of ROS can yield both double- and single-strand damage, depending on the spatial distribution of the energy deposition/ROS actions This paragraph and figure need to present these facts more accurately.

Response:

We have thoroughly reviewed this information and corrected the text as necessary. Additionally, we have included appropriate citations and updated Figure 1 to accurately reflect these corrections (lines 129-134).

9.- Line 147: SSBs can be effectively repaired by base excision repair (BER), nucleotide excision repair (NER), and DNA mismatch repair (MMR) mechanisms.

This is technically incorrect - SSBs are indeed created by these repair systems, so in the end SSBs are managed by these repair systems. However, SSBs themselves can be simply repaired by DNA ligation.

Response:

In the initial draft of the manuscript, lines 139-140, We discussed this aspect of repair systems, focusing mainly on the BER repair system. Furthermore, we have included additional evidence regarding the roles of the BER, NER, and MMR repair systems in addressing SSBs and other DNA lesions (lines 140-143).

10.- Line 157: IR doesn't cause premature chromosome condensation - PCC is instead a technique used to identify chromosomal aberrations after IR.

Response:

We apologize for any confusion. To enhance clarity, we have made the following corrections to the text: "At the cellular level, the radiotherapeutic treatment causes morphological changes in the cell and multiple damages in subcellular structures and organelles, such as reorganization of the cytoskeleton, promoting changes in cell surface micromorphology, mitochondria, and endoplasmic reticulum" (lines 149-153)

11. - Line 201: unclear as written. Rephrase as: ...through modulation of mechanisms such as autophagy...

Response:

We have revised the paragraph, aiming to enhance clarity. We hope this version is more precise (lines 196-199).

12.- Line 229: IR-induced (?)

Response:

Thank you for the observation. We have corrected the spelling of the word (line 226).

13.- Line 288: ...CircVRK1, an EMT-mediated signaling molecule.

Response:

Thank you. In the new version of the manuscript, we have clarified this term (lines 286-287).

14.- Line 301: is there a more up-to-date reference for acquired resistance than ref 68 (from 1967)?

Response:

We have updated the mentioned reference to ensure the information is clear (line 299).

15.- Lines 345-350: this complex sentence seems to be missing a phrase - note the unpaired parenthesis in the middle.

Response:

We apologize for any confusion. The error has been rectified in the latest version of the manuscript (lines 346 – 348).

16.- Line 353: ...resistance mechanism...

Response:

To enhance clarity, we have included the following phrase in the paragraph: “Another mechanism related to radio-resistance” (lines 352-353).

17.- Line 385-387: a reference supporting this statement should be added.

Response:

In this revised manuscript version, we have included the previously missing references (line 386).

18.- Line 426: do you mean ionizing radiation instead of infrared?

Response:

The accurate term is 'ionizing radiation.' This term has been rectified in the updated version of the manuscript (line 426).

19.- Line 434: RT may have systemic effects, but it is highly targeted so it is odd to describe as a systemic treatment, like chemotherapy.

Response:

To eliminate any potential confusion, we have revised the wording of the paragraph in question, and the error has been rectified in the latest version of the manuscript (lines 434-435).

20.- Line 524: Rephrase: In vivo knock-down of CCAR1 after irradiation....

Response:

Thanks for your suggestion. We have rewritten this paragraph to make it more straightforward (lines 525-528)

21.- Line 534: radio-responsive miRNAs

Response:

The modification has been incorporated into the latest version of the manuscript (line 537).

22.- Line 569-570: should be written as ATK and RAD52 instead of AKT/RAD52 as the slash designation usually indicates two names for the same molecule or gene.

Response:

Thanks for your suggestion. The information has been reformulated and included in supplementary table 1

Point-by-point response to Reviewer 2's comments.

1.- First, I found that, as written, the linkage and rationale between radiotherapy and miRNAs was somewhat brief and not necessarily well-connected. I understand the interest of the authors in this topic, but think that it would be useful to lay out the rationale more explicitly and clearly. This could be done by reframing the abstract and introduction, pointing out the "big picture" means by which miRNAs could modulate responses to radiotherapy, and building on these major themes throughout the rest of the review. This is done to some extent, but it needs to be more systematic. For example, lines 116-121 and 163-166 are the rationale/linkage between radiotherapy and miRNAs, but they are somewhat weak. For instance, could we not also look at transcription factors or some other class of gene expression regulators-I wanted to know why miRNAs, specifically, in radiotherapy?

Response:

Thank you for your valuable feedback. In the revised version of the manuscript, we have endeavored to explicitly illustrate the connection between the regulation of genetic expression mediated by miRNAs' regulatory function and the modulation of cellular processes associated with the radioresistant phenotype. Initially, our focus was solely on exploring the role of miRNAs in the development of radioresistance, aiming to precisely define the study's scope. By concentrating on this specific regulatory mechanism, we were able to delve into a broader spectrum of studied molecules, cancer types, and resultant phenotypes.

2.- Second, the abstract and review are wordy and could be streamlined. One way to do this would be to break it up into multiple reviews (miRNAs as markers is somewhat a separate topic). Another way is to simply streamline the writing style to make it more succinct-a lot of the early part of the review is dedicated to radiotherapy cellular

responses, which is necessary, but could be shortened to get into the discussion of miRNAs (consistent with the title of the review) at an earlier point in the review.

3.- Third, the figures very nicely summarize much of the writing in a more intuitive way. In contrast, in many sections the text read like a laundry list of miRNAs, with a one-sentence summary of what a miR does in cancer tissue/type X/Y/Z. I found the format of statements like "miR A does X in tissue Y" difficult to keep in focus and really absorb as I read, and would have benefitted from having a more unified discussion that placed the biological role at the forefront and then detailed related miRNAs or related cancers or used some other common biology thread to link the discussion. This was done to some extent in the molecular mechanisms section and subsections, and I would have benefitted from more of this type of organization throughout the review.

Response to 2 and 3:

Thank you for the suggestions. We've revised portions of the article, emphasizing conciseness and streamlining sections to enhance readability. In line with point 3, we've reorganized sections discussing numerous examples of miRNAs and their roles in specific cancer types. In the updated manuscript, we've incorporated an interactive table for intuitive visualization of this information, encompassing 157 miRNAs across 38 biological scenarios associated with 22 radiotherapy responses discussed in 165 articles.

4.- Line 115: delete "that provide helpful information"

Response:

Thank you for the suggestion; the phrase has been omitted in the revised version of the manuscript.

5.- Line 154 "multiple damages in" should be "multiple types of damage."

Response:

The modification has been implemented in the latest version of the manuscript (line 150).

6.- Line 161: clarify the difference between multiple cell nuclei and cells with numerous micronuclei

Response:

To prevent any potential confusion, we have revised the wording of this paragraph to enhance clarity (line 156)

7.- Line 210: "sublines" should be "sub-lines"

Response:

Thank you for your input. The modification has been incorporated into the latest version of the manuscript (line 206)

8.- Line 249: "down-modulated" should be "down-regulated"

Response:

Thank you. The modification has been incorporated into the latest version of the manuscript (line 246).

Response:

9.- Line 265: "Cancer stem cells:" should be bold

Response:

Thanks for noticing; the modification has been incorporated into the latest version of the manuscript (line 263).

10.- Line 404 "disease-free" should be "disease-free time"

Response:

The modification has been implemented in the latest version of the manuscript (lines 404 - 405).

April 24, 2024

RE: GENETICS-2024-306990

Dear Dr. Pedroza-Torres:

I am pleased to accept your manuscript entitled "Radio-miRs: a comprehensive view of radioresistance-related microRNAs" for publication in GENETICS, pending minor revision. You should be able to address all of the issues raised by Reviewer 1 through edits. Both reviewers and I feel that review can be more concise. Please make an effort to shorten the text and eliminate repetition.

Please submit your revision along with a brief description of how you modified the manuscript in response to the reviewers' concerns and suggestions (which can be viewed at the bottom of this email). I expect you should be able to submit a revised manuscript within 30 days. A suitably revised manuscript will be acceptable for publication; I don't expect to send it out for review, but I will go over it again before formal acceptance.

Thank you for submitting this review to Genetics.

Sincerely,

Jeff Sekelsky
Senior Editor
GENETICS

Approved by:
Howard Lipshitz
Editor in Chief
GENETICS

Reviewer comments:

Reviewer #1 (Comments for the Authors (Required)):

GENETICS-2024-306990

Radio-miRs: a comprehensive view of radioresistance-related microRNAs

Pedroza-Torres et al.

This is a revision of a comprehensive overview of radio-miRs and their roles in radiotherapy response, as prognostic biomarkers, and as potential interventional tools.

In general the authors have adequately addressed my previous concerns and those of the other reviewers. However, a couple minor issues remain as noted below. Some of these issues may have arisen during revision; others I may have missed in my earlier reading.

In text citations with two authors (e.g., Smith and Smith) are shown as Smith y Smith (Spanish). This is likely due to an incorrect Endnote setting.

Line 131: this paragraph has been revised, but it still implies that direct IR damage causes DSBs, whereas indirect ROS causes DSBs and single-strand damage. It needs to be further revised to indicate that direct damage (energy absorption) and indirect damage (ROS) cause single-strand lesions like base damage and SSBs, as well as DSBs.

Lines 161-165 and elsewhere: The authors first state that miRNAs regulate IR signaling, but later in the same sentence 'microRNA' is spelled out. Then on line 165, miRNA abbreviation is defined as microRNAs. This and all other abbreviations need to be defined at first use, and spelled out versions (e.g., microRNA) should never be used after being defined. Another example: ROS is defined on lines 121 and 259; there may be other examples I didn't catch. I recommend searching for all abbreviations and fully spelled-out versions to ensure that they are defined only at first use and only abbreviations used thereafter. Another rule: don't use abbreviations unless the term is used several times (GZMB is defined on line 350, but is never used again, so rather than shortening the text, this abbreviation lengthens it and thus serves no purpose. These recommendations apply to the

figure legends as well: no need to define abbreviations in legends if they've already been defined in the text (e.g., Fig 1 legend).

Line 194: ...and WNT/ β -catenin...

Line 202: ...modulating various mechanisms and cell types... [the latter phrase is needed because CSCs are not a mechanism]

Line 214: ...which suppresses BECN1, in a radioresistant pancreatic cancer cell line...

Line 245: Recent evidence has shown that...

Line 265: ...antioxidant enzyme...

Line 295: ...that mediates EMT...

Line 301: ...result of radiation on cancer cells...

Line 325-6: ...combinatory treatment of radiotherapy (local effects) and immunotherapy (systemic effects), which in fact has been shown to improve therapeutic response.

Line 338: omit 'Overall'

Line 341: ...radiation...

Line 441: ...are needed to identify...

Line 449: ...late effects appear thereafter...

Line 467: ...miR-99a and miR-221 expressed in...

Line 546: Further studies are needed to identify miRNA biomarkers of radiation injury to minimize radiotoxicity.

Reviewer #2 (Comments for the Authors (Required)):

With all due respect to the authors, I still find the review to be verbose, and there remain many instances where the text could be streamlined. For example, lines 399-404 are a one sentence paragraph talking about the subsequent section. I'm not sure what the purpose of this paragraph is for this section; it could be shortened and placed as the introductory sentence for the subsequent section, for instance. In other examples, the abstract remains quite long, and lines 110-115 basically repeat what is stated in the abstract and could be streamlined. There are more examples, but I won't point out all of them as this may be a style issue.

I also think the figures and tables provide a useful summary and valuable resource for their field, so I commend the authors on these features of the review.

Dear editors and reviewers,

I want to express my gratitude for taking the time to review our work and for providing valuable suggestions for improvement.

We have carefully implemented all the changes recommended by both reviewers, paying special attention to the use of abbreviations throughout the text, as pointed out by reviewer 1, and ensuring the correct formatting of references.

We have diligently rectified all instances where the 'Smith y Smith' format was mistakenly retained and replaced them with the correct 'Smith and Smith' format.

Additionally, we have made significant efforts to condense the text and eliminate any unnecessary repetition throughout the manuscript.

I trust that this revised version of the manuscript will fulfill all the necessary requirements while upholding the quality and originality of our work.

May 28, 2024

RE: GENETICS-2024-306990R1

Dr. Abraham Pedroza-Torres
Instituto Nacional de Cancerologia
Clínica de Cancer Hereditario
Av. San Fernando 22, Col Seccion XVI, Tlalpan, Ciudad de Mexico,
Ciudad de Mexico, N/A 14410
Mexico

Dear Dr. Pedroza-Torres:

Congratulations, your review entitled "Radio-miRs: a comprehensive view of radioresistance-related microRNAs" is accepted for publication in GENETICS! Many thanks for contributing to GENETICS.

To Proceed to Publication:

1. Format your article according to GENETICS style: <https://academic.oup.com/genetics/pages/general-instructions>

2. Ensure that you comply with data and community resource citation guidelines:
<https://academic.oup.com/genetics/pages/general-instructions#Data-Policy>

3. Upload your final files at <https://genetics.msubmit.net>

4. Add oupsupport@scipris.com and genetics.oup@novatechset.com (or the domains @scipris.com and @novatechset.com) to your email program's "safe senders" list. You will be contacted by both at various points during the production process.

Notes:

- We invite you to submit an original color figure related to your paper for consideration as cover art. Please email your submission to the editorial office or upload it with your final files. You can submit a small-sized image for evaluation, and if selected, the final image must be a TIFF file 2513px wide by 3263px high (8.375 by 10.875 inches; resolution of 600ppi). Please avoid graphs and small type.

- After files are sent to Oxford University Press we use SciPris to manage article licensing and payment. If you do not have a SciPris account, you will receive an email from no-reply@scipris.com to sign up to use Oxford University Press' author portal. After logging in, follow the online instructions to sign your licence. It is important that you select the Standard License to Publish so that the GSA will be billed for the page charges (Open Access is not covered by the GSA).

If you have any questions or encounter any problems while uploading your accepted manuscript files, please email the editorial office at sourcefiles@thegsajournals.org.

Sincerely,

Jeff Sekelsky
Associate Editor
GENETICS

Approved by:
Howard Lipshitz
Editor in Chief
GENETICS